# Is self-screening for 'at risk of malnutrition' feasible in a home setting?

Randi J. Tangvik[1,2]*, Eli Skeie[3,4], Arvid Steinar Haugen[1,5], Stig Harthug[1,6], Kristin Harris[1,7]

1 Department of Anaesthesia and Intensive Care, Haukeland University Hospital, Bergen, Norway, 2 Department of Clinical Medicine, Centre for Nutrition, University of Bergen, Bergen, Norway, 3 Department of Research and Development, Haukeland University Hospital, Bergen, Norway, 4 Department of Health and Social Services, Kvam Municipality, Norheimsund, Norway, 5 Faculty of Health Sciences, Department of Nursing and Health Promotion Acute and Critical Illness, OsloMet–Oslo Metropolitan University, Oslo, Norway, 6 Department of Clinical Science, University of Bergen, Bergen, Norway, 7 Department of Health and Caring Sciences, Western Norway University of Applied Sciences, Bergen, Norway

* randi.tangvik@uib.no

**Data Availability Statement:** All relevant data are within the manuscript and its Supporting Information files.

**Funding:** ES received PhD-grant from the Western Norway Regional Health Authority Trust (grant

## Abstract

### Introduction

Despite malnutrition being established as a well-known risk for postoperative complications, the lack of screening for nutritional risk remains a challenge. The aim of this study was to investigate whether self-screening for nutritional risk prior to surgery is feasible in a home setting and if it will increase number of patients screened for nutritional risk, and secondly, to compare their screening results with the "in-hospital assessments" conducted by healthcare professionals.

### Materials and methods

This was a prospective study involving patients from six randomly selected surgical wards at two Norwegian hospitals as a part of the "Feasibility study of implementing the surgical Patient Safety Checklist the (PASC)". This checklist included a self-reported screening tool based on the Nutritional Risk Screening tool (NRS 2002) to identify "at risk of malnutrition" in patients that will undergo surgery the next 3 months or less. The original screening tool (NRS 2002) was used as a standard routine to identify "at risk of malnutrition" by healthcare professionals at hospital. The interrater reliability between these results was investigated using Fleiss multi rater Kappa with overall agreement and reported with Landis and Koch's grading system (*poor*, *slight*, *fair*, *moderate*, *substantial*, and *almost perfect*).

### Results

Out of 215 surgical patients in the home setting, 164 (76.7%) patients completed the self-reported screening tool. A total of 123 (57.2%) patients were screened in-hospital, of whom 96 (44.7%) prior to surgery and 96 (44.7%) were screened both at hospital (pre- and post-surgery) and at home. Self-screening at home improved malnutrition screening participation by 71.9% compared to hospital screening prior to surgery (165 (76.7%) and 96 (44.7%),

number 912214). ASH received research grant from the Western Norway Regional Health Authority Trust (grant number HV1172). https://helse-vest.no/en KH received PhD grant from the Western Norway University of Applied Sciences (no grant number). https://www.hvl.no/en/ The funders did not play any role in the study design, data collection and analysis, decision to publish, or preparation of the manuscript.

**Competing interests:** I have read the journal's policy and the authors of this manuscript have the following competing interests: ASH has previously received travel re-imbursement for representing the International Federation of Nurse Anaesthetists in the European Society of Anaesthesiology and Intensive Care's Patient Quality and Safety Committee. All the other authors have declared that no competing interests exist. This does not alter our adherence to PLOS ONE policies on sharing data and materials.

respectively) and by 34.1% compared to pre- and postoperative in-hospital screening, 165 (76.7%) and 123 (57.2%), respectively). The degree of agreement between patients identified to be "at risk of malnutrition" by the self-reported screening tool and healthcare professionals was *poor* ($\kappa$ = - 0.04 (95% CI: -0.24, 0.16), however, the degrees of agreement between the patients and healthcare professionals answers to the initial NRS 2002 questions "low BMI", "weight loss", and "reduced food intake" were *almost perfect* ($\kappa$ = 1.00 (95% CI: 0,82, 1.18)), *moderate* ($\kappa$ = 0.55 (95% CI: 0.34, 0.75)), and *slight* ($\kappa$ = 0.08 (95% CI: - 0.10, 0.25) respectively.

## Conclusions

Three out of four patients completed the self-screening form and the preoperative screening rate improved with 70%. Preoperatively self-screening in a home setting may be a feasible method to increase the number of elective surgical patients screened for risk of malnutrition.

## Trial registration

The trial is registered in ClinicalTrials.gov ID NCT03105713. https://classic.clinicaltrials.gov/ct2/show/NCT03105713.

## Introduction

Malnutrition is an established risk factor for postoperative complications and should be prevented and treated prior to surgery [1,2]. To identify patients who are diagnosed as malnourished or "at risk of malnutrition", both European [3] and Norwegian [4] guidelines for preventing and treating malnutrition recommend nutritional risk screening at admission. For this purpose, the screening tool Nutrition Risk Screening 2002 (NRS 2002) [5] has been recommended for use by healthcare professionals for assessing in-hospital surgical patients [3,6]. The nutritional risk screening and associated treatment should occur early in the patient care process [6], since the risk of malnutrition may worsen over time and negatively affect patient outcomes [7]. Regarding elective surgery, the time between when surgery is booked, and the actual surgical procedure may be suitable for detecting and starting to treat malnutrition. The most recent recommendations for perioperative nutrition from the European Society for Clinical Nutrition and Metabolism expert group recommend that patients be screened for "at risk of malnutrition" at least 10 days before surgery [6]. However, this recommendation may be difficult to follow since most elective patients are admitted at hospital at a time closer to the surgery.

The World Health Organization and European patient organizations endorse patient's involvement in safety [8,9], and it has been suggested that patients can screen themselves for "at risk of malnutrition" [10]. Moreover, surgical patients and healthcare professionals have identified nutritional status to be a safety risk factor that should be included in a patient-driven surgical patient safety checklist [11]. Previous studies have demonstrated that patients can successfully screen themselves in an outpatient setting for "at risk of malnutrition" using the Patient-Generated Subjective Global Assessment Short Form [12], the Malnutrition Universal Screening Tool (MUST) [10] and the Malnutrition Screening Tool (MST) [13]. However, there is a lack of knowledge and a need for studies investigating if elective surgical patients can screen themselves for "at risk of malnutrition" in their home setting. The aim of the study was to investigate whether it is feasible for patients to answer questions related to nutritional risk

screening in a home setting prior to hospital admission for surgery, and to investigate whether this could increase the number of patients being screened for risk of malnutrition. A secondary aim was to assess the agreement between patients' perception of being at risk of malnutrition and healthcare professionals' screening, using the NRS 2002 screening tool.

## Material and methods

### Study design

This prospective trial is part of the "Feasibility study of the Patient Safety Checklist (PASC)" to be used by surgical patients before and after surgery, prior to a stepped wedge cluster randomized controlled trial (SW-CRCT) [14].

The power calculation was based on the PASC trail protocol. It was calculated that the present study would require a total of 300 elective surgical patients, across six clusters, over 14 months [14].

### Settings and participants

Study participants were recruited in the period between August 2019 through September 2020 from Haukeland University Hospital and Førde Central Hospital in Western Norway. Six surgical specialties were randomly drawn from eleven eligible surgical wards. All wards agreed to participate: Ear, Nose, Throat (ENT)-/Maxillo-Facial; Cardio-Thoracic; Neuro; Breast- and Endocrine; Gastrointestinal-; and General surgery. The study participants were recruited by study personnel in cooperation with the nurses and surgeons at each wards Outpatients Clinique or by letter in the mail, after the patient had received a date for surgery. Inclusion criteria for the participants were: elective surgical patients aged 18 years or older, cognitively capable of answering the PASC checklist including nutritional risk screening questions, living at home, able to give informed consent and fluent in Norwegian. Participants were prospectively recruited when scheduled to surgery by the hospital within a period of 2 to 12 weeks before surgery, depending on the patients' severity of disease and need for surgery. Patients returned their completed PASC checklists, including nutritional risk screening questions, before discharge or by mail. Flowchart of the study population is shown in Fig 1.

### Assessment of nutritional status

According to the regional and the current national guidelines [15,16], a Norwegian translated version of NRS 2002 [17] embedded in the electronic patient record was used by healthcare professionals at the hospitals, mainly nurses, to determine whether the patients were at "risk of malnutrition" or not. This screening tool is based on four introductory questions that detect low body mass index (BMI) ($< 20.5 \text{ kg/m}^2$), recent weight loss, recently reduced food intake and critical illness [5]. If one or more of these four questions are answered with "yes", the patient enters the final screening. The final screening gives a total score from 0 to 7 based on more in-depth questions regarding the patient's nutritional status (score 0–3) and the severity of the patient's disease in consideration of nutritional requirements (score 0–3). In addition, patients over 70 years are given the score 1. A total score of $\geq 3$ in the final screening identifies patients as "at risk of malnutrition" [3]. NRS 2002 is designed for use by healthcare personnel. Some adjustments were made for patients' self-screening with NRS 2002; see **Fig 2**. First, the initial question regarding critical illness and the final screening questions that quantify the severity of their disease to be moderate or severe were left out, since patients in such cases would not be at home but hospitalized. Secondly, the screening tool was re-design to be more patient user-friendly. Finally, based on the patients' answers to the questions and their age,

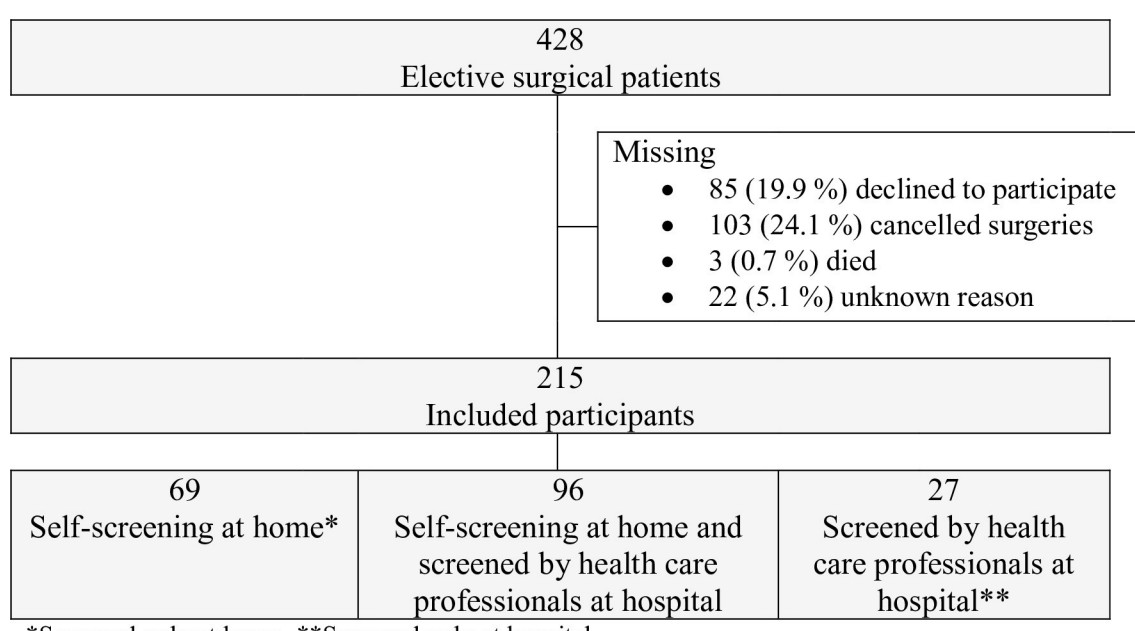

**Fig 1. Flowchart of the study population.**

study personnel calculated the BMI and, NRS 2002 scores, and categorized the patients to be "at risk of malnutrition" ($\geq$ 3 scores) or not ($<$ 3 scores).

Self-registration of the nutritional information as part of the PASC [14], is hereby called the "self-reported screening tool" and was performed at home, up to 3 months prior to surgery.

---

**Self-reported screening tool for use at home**

**Height**
How tall are you now?
Some lose height with age. What is the tallest you have been measured?

**Weight**
What is your body weight now?

What was your body weight
 1 month ago?
 2 months ago?
 3 months ago?
 6 months ago?

**Food intake**
Have you eaten less than usual in the last two weeks?
How much have you eaten last week, compared to what is normal for you?
 As usual or more?
 Slightly more than half of normal?
 Half to a quarter of normal?
 Less than a quarter of normal?

**Disease**
Do you have a chronic disease e.g., liver cirrhosis, COPD, kidney failure, diabetes, cancer or other? Y/N

**Malnutrition**
Do you think you are malnourished or at risk of becoming so? Y/N

---

**Nutritional Risk Screening 2002 (NRS 2002) for use at hospital**

**Initial screening**

| | |
|---|---|
| Is the BMI of the patient < 20.5 kg/m²? | **Y/N** |
| Did the patient lose weight in the past 3 months? | **Y/N** |
| Was the patient's food intake reduced in the past week? | **Y/N** |
| Is the patient critically ill? | **Y/N** |

<u>If yes</u> to one of those questions, proceed to main screening.

<u>If no</u> for all answers, the patient should be re-screened weekly.

| **Main screening** | **Score** |
|---|---|
| **Nutritional status** | **0-3** |
| None | 0 |
| Mild: Weight loss >5% in 3 months OR 50–75% of the normal food intake in the last week | 1 |
| Moderate: Weight loss >5% in 2 months OR BMI 18.5–20.5 kg/m2 AND reduced general condition OR 25–50% of the normal food intake in the last week | 2 |
| Severe: Weight loss >5% in 1 month | 3 |
| **Severity of the disease** (stress metabolism) | **0-3** |
| **None** | 0 |
| **Mild:** Patient is mobile. Increased protein requirement can be covered with oral nutrition. Hip fracture, chronic disease especially with complications e.g., liver cirrhosis, COPD, diabetes, cancer, chronic hemodialysis | 1 |
| **Moderate:** Patient is bedridden due to illness. Highly increased protein requirement may be covered with ONS. E.g., stroke, hematologic cancer, severe pneumonia, extended abdominal surgery. | 2 |
| **Severe: E.g., i**ntensive care patient, major surgery. | 3 |
| **Age** | **0-1** |
| ≥70 years | 1 |

---

**Fig 2. Screening for "at risk of malnutrition".**

To raise awareness of nutritional status prior to surgery, the patients were asked on the same paper form if they thought they were "at risk of malnutrition" ("yes"/"no"). These results were compared with the healthcare professionals' nutritional risk screening results which was performed at the hospital 2 weeks before and until the day before surgery and in some cases after surgery.

The screening process is illustrated in **Fig 3**.

## Statistics

Descriptive analyses were conducted for the total study sample and for the subsamples attributed to different surgical wards. Summary measures for continuous variables are reported as means (SD), and categorical variables are reported as counts (percentages). The statistical package IBM SPSS Statistics v.26 was applied for descriptive analysis. To analyse the interrater reliability between the patients' and the healthcare professionals' results of the nutritional risk screening, the Fleiss multi rater Kappa Measure of Agreement (κ) was investigated by using SPSS (version 26) [18]. The grading system of Landis and Koch was used to describe the agreement (< 0.00, *poor*; 0.00–0.20, *slight*; 0.21–0.40, *fair*; 0.41–0.60, *moderate*; 0.61–0.80, *substantial*; 0.81–1.00, *almost perfect*) [19,20].

## Ethics

The study was approved by the Regional Committee for Medical and Health Research Ethics prior to study start (Reference: 2016/1102) and the hospital managers. All study participants signed a written consent after receiving information regarding the study. Participation was

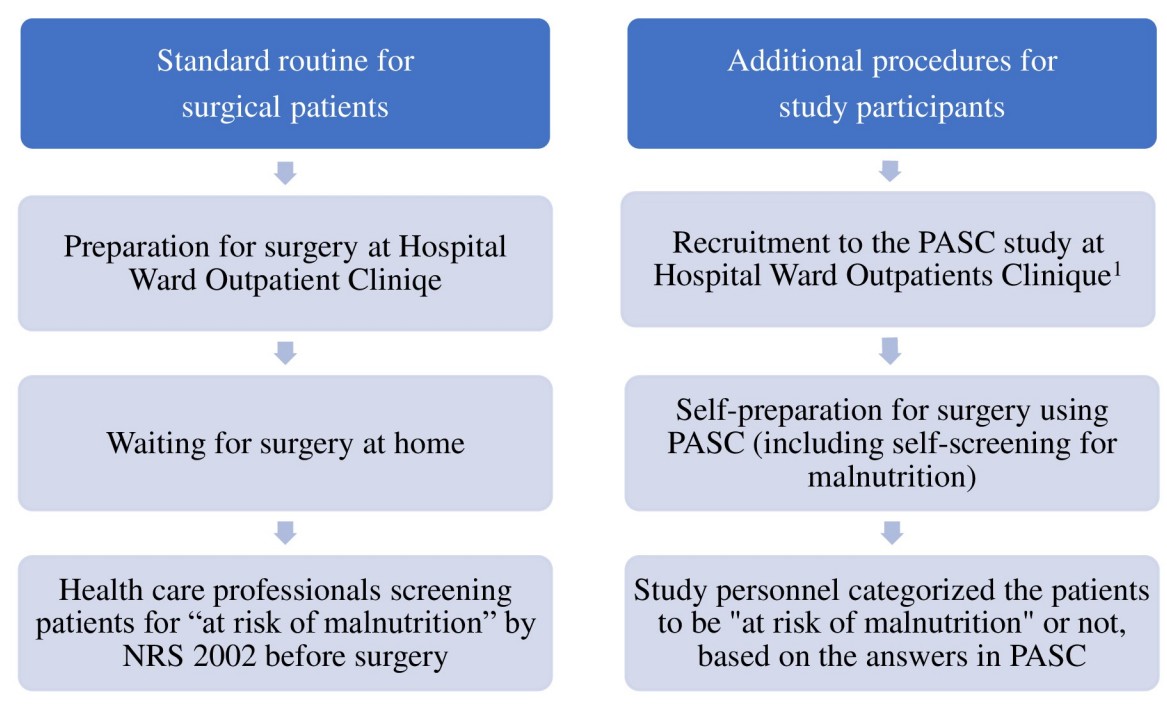

Abbreviations: PASC Patients surgery checklist.
[1] A few study participants were recruited by letter in the mail after receiving a date for surgery.

**Fig 3. Illustration of the malnutrition screening process in the present study.** PASC: Patient Safety Checklist trial & NRS 2002: Nutritional Risk Screening 2002 [17].

voluntary and patients could withdraw at any time without consequences. Research data were handled according to the hospital's research guidelines (eProtocol: 1218–1218) and data were stored on a designated and secure research server in one of the hospitals (S1 Table).

## Results

### General characteristics

Out of 428 patients asked to participate, 215 (50.2%) patients consented, used, and returned the PASC where the self-reported screening tool was included (**Fig 1**). Of these, 100 (46.5%) were male, and the mean (SD) age was 57.8 (13.0) years. General characteristics of the study participants and distribution among the different surgical wards are described in **Table 1**.

### Screening performance

At home and using the self-reported screening tool, 164 (76.7%) patients completed enough of the questions to be classified as "at risk of malnutrition" or not. In total, 205 patients (95.3%) answered "yes" or "no" to the question regarding whether they thought they were "at risk of malnutrition" prior to surgery. For the initial questions regarding the patients' BMI, weight loss and dietary intake in NRS 2002 in the self-reported screening tool, the response rates were 92.1%, 77.7% and 95.3%, respectively.

At the hospital, healthcare professionals completely screened 123 (57.2%) out of the 215 patients for "at risk of malnutrition" or not, of which 96 patients (44.7%) were screened preoperatively. Twenty-seven (12.6%) of these patients did not complete the self-screening. The healthcare professionals' response rates for the initial questions regarding the patients' BMI, weight loss and dietary intake in NRS 2002 were 59.5%, 59.1% and 59.5%, respectively. Self-screening improved malnutrition screening participation by 71.9% compared to hospital screening prior to surgery (165 (76.7%) and 96 (44.7%), respectively) and by 34.1% compared to pre- and postoperative screening at the hospital 165 (76.7%) and 123 (57.2%), respectively).

**Table 1. Distribution of patient characteristics and screening performance of the total study sample and subsamples.**

| | Total n = 215 | Haukeland University Hospital | | | | | Førde Central Hospital |
| --- | --- | --- | --- | --- | --- | --- | --- |
| | | Gastrointestinal n = 37 | Breast and endocrine n = 45 | ENT/ Maxillo-Facial n = 43 | Neuro n = 32 | Cardio-Thoracic n = 36 | General n = 22 |
| Age,[1] | 57.8 (13.0) | 59.0 (12.8) | 59.8 (9.8) | 50.0 (15.7) | 54.5 (9.6) | 62.7 (10.0) | 64.1 (14.7) |
| Male[2] | 100 (46.5) | 18 (48.6) | 2 (4.4) | 21 (48.8) | 15 (46.9) | 30 (83.3) | 15 (63.6) |
| *At hospital*[2] | | | | | | | |
| Complete screening | 123 (57.2) | 27 (73.0) | 1 (2.2) | 22 (51.2) | 31 (96.9) | 33 (94.3) | 8 (36.4) |
| "At risk of malnutrition" | 4 (1.9) | 4 (10.8) | 0 | 0 | 0 | 0 | 0 |
| *At home*[2] | | | | | | | |
| Complete self-screening | 164 (76.7) | 32 (86.5) | 31 (68.9) | 36 (83.7) | 23 (71.9) | 25 (71.4) | 17 (77.3) |
| "At risk of malnutrition" | 11 (5.1) | 3 (8.1) | 1 (2.2) | 3 (7.0) | 0 | 1 (2.8) | 3 (13.6) |
| Believe to be "at risk of malnutrition" | 9 (4.2) | 3 (8.1) | 3 (6.7) | 1 (2.3) | 0 | 0 | 2 (9.1) |

Abbreviations: ENT: Ear, Nose, Throat. [1]mean (SD), [2]n (%).

"At risk of malnutrition" is defined by the Nutritional Risk Screening 2002 [5].

## "At risk of malnutrition"

According to the answers given in the self-reported screening tool preoperatively, 11 patients (5.1%) met the criteria for being "at risk of malnutrition" and nine patients (4.2%) believed they were in this category. At the hospital, four of the 215 patients (1.9%) were identified to be "at risk of malnutrition" by the healthcare professionals' screening. All these patients were screened after surgery.

## Agreement regarding the initial NRS 2002 questions

Out of 164 (76.7%) patients completing the self-reported screening tool at home, 96 (67.9%) were screened pre- or postoperatively by a healthcare professional at hospital. The degrees of agreement between the answers of patients and healthcare professionals to the initial NRS 2002 questions were as follows: low BMI: *almost perfect* ($\kappa$ = 1.00 (95% CI: 0,82, 1.18)), weight loss: *moderate* ($\kappa$ = 0.55 (95% CI: 0.34, 0.75)), and reduced food intake: *slight* ($\kappa$ = 0.08 (95% CI: - 0.10, 0.25)) (Table 2).

## Agreement regarding being "at risk of malnutrition" or not

Comparing patients identified to be "at risk of malnutrition" by healthcare professionals with the self-reported screening tool and the patients' belief regarding their own nutritional status, the degree of agreement was demonstrated to be *poor* for them both ($\kappa$ = - 0.04 (95% CI: -0.24, 0.16) and $\kappa$ = - 0.03 (95% CI: - 0.21, -0.15), respectively) (Table 3).

**Table 2. Degrees of agreement regarding the initial NRS 2002 questions [5] between patients and healthcare professionals.**

| | Initial NRS 2002 questions | Healthcare professionals | | | | Kappa (95% CI) [1] | Grading of agreement [2] |
|---|---|---|---|---|---|---|---|
| | BMI < 20.5 kg/m$^2$? | *Yes* | *No* | *Missing* | *Total* | | |
| **Patients** | *Yes* | 4 | 0 | 7 | 11 | 1.00 (0.82, 1.18) | Almost perfect |
| | *No* | 0 | 113 | 73 | 186 | | |
| | *Missing* | 0 | 11 | 7 | 18 | | |
| | *Total* | 4 | 124 | 87 | 215 | | |
| | | **Healthcare professionals** | | | | | |
| | Weight loss last three months? | *Yes* | *No* | *Missing* | *Total* | 0.54 (0.34, 0.75) | Moderate |
| **Patients** | *Yes* | 11 | 12 | 25 | 48 | | |
| | *No* | 3 | 76 | 40 | 119 | | |
| | *Missing* | 0 | 25 | 23 | 48 | | |
| | *Total* | 14 | 113 | 88 | 215 | | |
| | | **Healthcare professionals** | | | | | |
| | Reduced food intake last week? | *Yes* | *No* | *Missing* | *Total* | 0.08 (-0.10, 0.25) | Slight |
| **Patients** | *Yes* | 3 | 17 | 16 | 36 | | |
| | *No* | 8 | 94 | 67 | 169 | | |
| | *Missing* | 0 | 6 | 4 | 10 | | |
| | *Total* | 11 | 117 | 87 | 215 | | |

**Abbreviations: NRS 2002 = Nutritional Risk Screening 2002.** [1] Fleiss multi rater Kappa with overall agreement. [2] The grading system of Landis and Koch was used to describe the agreement.

**Table 3. Degrees of agreement regarding "at risk of malnutrition" between the patients and healthcare professionals.**

| | | | "At risk of malnutrition" *screened by healthcare professionals* | | | | | Kappa (95% CI) [1] | Grading of agreement [2] |
|---|---|---|---|---|---|---|---|---|---|
| | | | *Yes* | *No* | *Missing* | | *Total* | -0.043 (-0.24, -0.16) | Poor |
| "At risk of malnutrition" | Patient's self-screening | *Yes* | 0 | 5 | 6 | | 11 | | |
| | | *No* | 3 | 88 | 63 | | 154 | | |
| | | *Missing* | 1 | 26 | 23 | | 50 | | |
| | | | | | | | | | |
| | | *Total* | 4 | 119 | 92 | | 215 | | |
| | | | | | | | | | |
| | | | "At risk of malnutrition" *screened by healthcare professionals* | | | | | | |
| | | | *Yes* | *No* | *Missing* | | *Total* | -0.031 (-0.21, -0.15) | Poor |
| | Patient's belief | *Yes* | 0 | 3 | 6 | | 9 | | |
| | | *No* | 4 | 109 | 83 | | 196 | | |
| | | *Missing* | 0 | 7 | 3 | | 10 | | |
| | | | | | | | | | |
| | | *Total* | 4 | 119 | 92 | | 215 | | |

[1] Fleiss multi rater Kappa with overall agreement

[2] The grading system of Landis and Koch was used to describe the agreement.

"At risk of malnutrition" is defined by the Nutritional Risk Screening 2002 (NRS 2002) [5]. Patient's self-screening based on answers given to NRS 2002 questions in paper form prior to admission.

The agreement between the self-reported screening tool, and the patients' belief regarding being "at risk of malnutrition" was *fair* (κ = 0.27 (95% CI: -0.12, 0.30)) (Table 4).

## Discussion

This study highlights the importance of self-screening at home and demonstrates that self-screening is possible and increases numbers of patients screened for malnutrition risk prior to surgery. Three out of four patients completed the self-screening for nutritional risk. The number of patients self-screening at home, preoperatively was more than 70% higher than in-hospital preoperative screening, and 34% higher than total pre- and postoperative in-hospital screening.

We found that most patients who met the criteria for being "at risk of malnutrition" in the self-screening tool and those who believed they were "at risk of malnutrition" preoperatively, were not identified to be so by the healthcare professionals at hospital. As compared to previous studies, this study has a lower prevalence of patients at risk of malnutrition, which may be due to a low number of patients screened in hospital [15]. In most cases, this was due to missing screenings at hospital, and not necessarily due to disagreement between the patients and the healthcare professionals. The agreement between the patients and healthcare workers regarding the initial NRS 2002-questions; low body mass index, weight loss and reduced food intake was *almost perfect*, *moderate*, and *slight*, respectively. However, the over-all agreement regarding classification of being "At risk of malnutrition" or not was *poor*. Therefore, NRS 2002 may not be suitable as self-screening tool for patients.

**Table 4. Degrees of agreement between patients' self-reported screening and their belief regarding being "at risk of malnutrition".**

| | | Patients believe they are "at risk of malnutrition" | | | | Kappa[1] (95% CI) | Grading of agreement [2] |
|---|---|---|---|---|---|---|---|
| | | Yes | No | Missing | Total | 0.27 (-0.12, 0.3) | |
| Self-screened to be "at risk of malnutrition" [3] | Yes | 3 | 8 | 0 | 11 | | Fair |
| | No | 5 | 147 | 2 | 154 | | |
| | Missing | 1 | 41 | 8 | 50 | | |
| | Total | 9 | 196 | 10 | 215 | | |

[1] Fleiss multi rater Kappa with overall agreement

[2]The grading system of Landis and Koch was used to describe the agreement.

[3] Patient's self-screening based on answers given to Nutritional Risk Screening 2002 [5] in paper form prior to admission.

The problem of missing screening for "at risk of malnutrition" is well-known both nationally and internationally [21,22]. When patients are not identified as being "at risk of malnutrition", it is difficult to further prevent and treat the disease-related malnutrition. In this way, the chance of reducing in-hospital mortality and readmissions decreases [23]. The current study demonstrated that patients tended to complete the questions needed to fulfil NRS 2002 in a home setting more often (76.7%) than what was obtained by healthcare professionals during hospitalization (57.2%). This is an important finding, since it illustrates a willingness and ability to perform nutritional self-screening in the patient group, which potentially could result in an increased rate of nutritional screening. User involvement has also previously been recognized as an important factor for successful patient treatment, and healthcare professionals are encouraged to increase user involvement in all aspects of their work, both within clinical work and research [8,24].

Early intervention in vulnerable groups to prevent malnutrition was voted as the most prioritized research question within the field of adult malnutrition and nutritional screening in healthcare by a partnership between patients, careers, and healthcare professionals [25]. Several factors may affect the patient's appetite before surgery, such as the underlying cause of surgery, as well as fear of surgery and/or pain or illness. This may further lead to a suboptimal nutritional status prior to surgery, and thus increase the risk of postoperative complications [7]. To be able to treat patients who are "at risk of malnutrition" prior to surgery, early screening must be conducted. Here, an organizational challenge occurs, since the patient responsibility lies between the primary and secondary healthcare system in the preoperative phase, and the responsibility for nutritional screening and treatment may become unclear. Thus, questions regarding risk of malnutrition are an important part of the patients' checklist prior to surgery.

For hospitals, the Norwegian guidelines for preventing and treating malnutrition recommend patients be screened for "at risk of malnutrition" within 24 hours after admission [4]. Self-screening at home will not change the hospitals responsibility for following these

guidelines. The healthcare personnel are trained for screening and are responsible for taking care of the patients' risk by assessment, nutritional treatment and finally to communicate information regarding nutritional therapy to next level of care. Of note, the organization of healthcare services in Norway has undergone some major changes during the last two decades, which has led to a reduced length of stay at hospital. As a result, many patients, who previously could be screened and identified for "at risk of malnutrition" at hospital within 24 hours, are now discharged. This leads to a lower chance of preventing malnutrition, as the patients "at risk of malnutrition" will not be identified, and thereby treated, during their hospital stay. The low prevalence of patients staying overnight at the ward for Breast and Endocrine surgery may explain why this ward has such a low screening performance (2.2%) among the patients included in the current study (see Table 1). Moreover, a significant number of these patients are having surgery for breast cancer. Since this patient group is known to have problems related to unintended weight gain in the postoperative phase [26], the focus on malnutrition may be decreased. Notably, 6.7% of the study participants at this ward thought they were "at risk of malnutrition" but were not screened for this during hospitalization.

The degree of agreement between patients who identified themselves as being "at risk of malnutrition" by the self-reporting screening tool and by personal belief, with the result from the screening at the hospital, was *poor*. This was partly due to high prevalence indexes, since the proportion of agreements on the positive classification (*poor*) differed from that of the negative classification (*almost perfect*) (data not shown). Of note, more patients were not screened for "at risk of malnutrition" at hospital (missing data) than there were cases of disagreement in the screening results.

None of the patients who were identified at hospital to be "at risk of malnutrition" by the healthcare professionals were identified to be so by the questions in the self-reporting screening tool, and none of them thought they were so prior to admission. These patients were screened by the healthcare professionals after the surgery. Thus, the *poor* degree of agreement may be caused by the time difference, since we know that surgery and hospitalization may lead to loss of appetite, and reduced dietary intake and body weight [7]. Notably, this illustrates the importance of optimal nutrition reserves prior to surgery, and at least weekly screening routines at the hospital [27], since the nutritional status may worsen in a short time.

In an attempt to overcome the barriers of low screening performance for "at risk of malnutrition", the Norwegian Directorate of Health recently published new guidelines for preventing and treating malnutrition [4]. In these revised guidelines, there was a strong consensus towards implementing MST [28] as a new screening tool for both the primary and secondary healthcare system. MST consist of two questions: one regarding weight changes and one regarding changes in dietary intake and is therefore easier to use than NRS 2002. Since the current study demonstrated patients were willing to screen themselves prior to elective surgery, and a previous study has reported patients successfully screened themselves with MST in an outpatient setting [13], there is reason to believe that MST can also be used in a home setting for self-screening. Therefore, it is important to facilitate a treatment offer for those who identify themselves as "at risk of malnutrition".

## Strengths and limitations

The strength of this study is the prospective design and the use of the daily nutritional risk screening routines at six different surgical wards from two hospitals. The main limitation is the low number of patients identified to be "at risk of malnutrition" at home, and at hospital, since this decreases the possibility of a reliable comparison of the screening results between patients and healthcare professionals. Also, due to the hospital's routines, in some cases it

could have been up to three months between the self-screening performed at home and the routinely screening performed at hospital. During the weeks prior to surgery, patients may experience weight loss and changes in their food intake that will impact upon the validity of the correlation statistics. Previously published analysis showed no difference between genders on responding, but demonstrated that the central community hospital had a higher number of non-responders, possibly indicating that patients with more complex surgery and medical conditions were the ones who participated in the study [14]. In addition, the modifications to NRS 2002 to customize the screening tool for patients' self-screening may have affected its reliability. Of note, similar modifications have also been suggested to self-reporting screening by using MUST [10]. A further limitation of the study is that healthcare professionals may screen the patients after surgery, whereas the patients screened themselves prior to surgery, and so we cannot exclude the possibility of a screening bias.

## Clinical relevance

This study demonstrated potential for elective patients to answer questions that can be used to assess their nutritional status prior to surgery in a home setting. Further development may allow for the identification of those patients who are "at risk of malnutrition" prior to hospitalization and utilize the waiting time prior to surgery to safeguard and improve their nutritional status. Further investigation should include the electronic formats that may improve the accuracy and ease of use of self-screening, and include an associated treatment offer for those who need it (S1 and S2 Files).

## Conclusion

In this study, three out of four surgical patients performed self-screening at home, thus the preoperative screening rate improved with 70%, which shows that patients are able to and willing to perform self-screening at home. This is an important step towards involving patients in their own surgical safety as patients self-screening is a feasible and important method to increase the number of elective surgical patients screened for risk of malnutrition before surgery. The next step is to investigate other self-screening tools and to give nutritional support to the patients that identify themselves with nutritional risk to improve nutritional status and thus patient safety prior to surgery.

## Supporting information

**S1 Table. An anonymized dataset.** Abbreviations: BMI: body mass index, FS: final screening, HP: healthcare professionals, IS: initial screening, NRS: Nutritional risk screening P: patients.
(PDF)

**S1 File. A CONSORT checklist.**
(PDF)

**S2 File. A copy of the study protocol at ClinicalTrails.** The current study was a part of a development and validation Patient Safety Checklist (PASC) (ClinicalTrails.gov: NCT03105713).
(PDF)

## Acknowledgments

We would like to express our thanks to all the patients who participated in the study and to the staff at the participating surgical wards for practical help during patient recruitment.

## Author Contributions

**Conceptualization:** Randi J. Tangvik, Eli Skeie, Arvid Steinar Haugen, Stig Harthug.

**Formal analysis:** Eli Skeie.

**Investigation:** Eli Skeie, Kristin Harris.

**Project administration:** Eli Skeie, Arvid Steinar Haugen, Kristin Harris.

**Supervision:** Randi J. Tangvik, Stig Harthug.

**Visualization:** Randi J. Tangvik.

**Writing – original draft:** Eli Skeie.

**Writing – review & editing:** Randi J. Tangvik, Eli Skeie, Arvid Steinar Haugen, Stig Harthug, Kristin Harris.

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
