## [Decision Letter · Decision Letter 0]

17 Mar 2023

PONE-D-22-30022Surgical patients’ self-screening for “at risk of malnutrition” in a home setting – a feasibility study (ClinicalTrails.gov: NCTT03105713)PLOS ONE

Dear Dr. Skeie,

Thank you for submitting your manuscript to PLOS ONE. After careful consideration, we feel that it has merit but does not fully meet PLOS ONE’s publication criteria as it currently stands. Therefore, we invite you to submit a revised version of the manuscript that addresses the points raised during the review process.

The reviewer has highlighted a critical area of confusion concerning the word “feasibility”. I suggest that the authors carefully examine their manuscript and locate all instances of “feasibility”, then replace it another word that conveys their meaning, such as “practicability”.

The other important point the reviewer makes is that study outcomes are not clear.

We look forward to receiving your revised manuscript.

Kind regards,

Maret G Traber, PhD

Academic Editor

PLOS ONE

Journal Requirements:

"I have read the journal's policy and the authors of this manuscript have the following competing interests: ASH has previously received travel re-imbursement for representing the International Federation of Nurse Anaesthetists in the European Society of Anaesthesiology and Intensive Care’s Patient Quality and Safety Committee. All the other authors have declared that no competing interests exist."

Additional Editor Comments:

The reviewer has highlighted a critical area of confusion concerning the word “feasibility”. Authors frequently are asked in clinical trials to state the kind of study design In the title. As noted “Feasibility studies are pieces of research done before a main study to answer the question 'Can this study be done? ' They are used to estimate important parameters that are needed to design the main study.”(see: https://www.nihr.ac.uk/glossary/?letter=F&postcategory=-1). In other words, a feasibility study is one that has not collected data, but has a study design and statistical support for that design. The study the authors have carried out might be better described as a “pilot trial” that shows positive results. To avoid the difficulty with the multiple meanings of feasibility, I suggest that the authors carefully examine their manuscript and locate all instances of “feasibility”, then replace it another word that conveys their meaning, such as “practicability”.

The other important point the reviewer makes is that they have not done a power calculation to estimate how many subjects they need to claim that the results are statistically significant. A power calculation may not be possible without the data they collected, suggesting another reason to call this a pilot trial.

Reviewers' comments:

Reviewer's Responses to Questions

**Comments to the Author**

1. Is the manuscript technically sound, and do the data support the conclusions?

Reviewer #1: Yes

2. Has the statistical analysis been performed appropriately and rigorously? 

Reviewer #1: Yes

3. Have the authors made all data underlying the findings in their manuscript fully available?

Reviewer #1: No

4. Is the manuscript presented in an intelligible fashion and written in standard English?

Reviewer #1: Yes

5. Review Comments to the Author

Reviewer #1: This study is presented as a feasibility study, however it doesn't seem to have actually been intended to assess feasibility of an intervention, as the objectives are presented as "investigating whether self-screening for nutritional risk factors prior to surgery in a home setting will increase the amount of patients screened for nutritional risk, and secondly, to compare their screening results with healthcare professionals..." (lines 26 to 29). Going by this, the objective of this study is to assess effectiveness of an intervention. The authors should be clearer about what the study was intending to achieve as the title, objectives in the abstract, and objectives at the end of the methods section are at odds.

If the authors did indeed aim to "investigate whether self screening... will increase the amount of patients screened...", then there are a number of aspects that need to be clearly reported, including the outcomes of the study and how they were measured, and also compared across groups. However, if the study aimed to investigate feasibility as stated in the title and in line 75, then the authors need to clearly describe what the feasibility endpoints are, ie. what outcome would they need to observe to conclude that self-screening was feasible. Additionally, the authors should describe how they determined the appropriate size of the study - a reader cannot tell whether the 215 patients who participated were sufficient to address the objectives of the study.

6. PLOS authors have the option to publish the peer review history of their article (what does this mean?). If published, this will include your full peer review and any attached files.

Reviewer #1: No

---

## [Author Response · Author response to Decision Letter 0]

29 Apr 2023

Dear Editor and Reviewer, 

I hereby submit the revised version of our manuscript “Practicability of self-screening for “at risk of malnutrition” among surgical patients’ in a home setting” (PONE-D-22-30022).

We thank the Editor and Reviewer for a thorough and thoughtful evaluation of the manuscript, as well as for critical and helpful comments. In our revision of the manuscript, we have addressed all the objections and suggestions raised to the best of our ability.

Regarding the Competing Interests section, we state as requested by the Journal Requirements the following: “This does not alter our adherence to PLOS ONE policies on sharing data and materials.”. The Journal Requirements also requested captions for our Supporting Information files which can now be found in the end of the manuscript. One of these also include an anonymized dataset. 

Below, you will find the point-by-point response to the Editor and Reviewer. All lines and pages referred to apply to the Revised Manuscript with Track Changes. We hope that our revised version has addressed their comments satisfactorily and is now at a level that it can be deemed acceptable for publication in PLOS ONE. 

We look forward to hear from you.

Best regards, 

Eli Skeie

Corresponding Author

 

Response to the Editor’s Comments

#1 The reviewer has highlighted a critical area of confusion concerning the word “feasibility”. Authors frequently are asked in clinical trials to state the kind of study design in the title. As noted “Feasibility studies are pieces of research done before a main study to answer the question 'Can this study be done? ' They are used to estimate important parameters that are needed to design the main study.”(see: https://www.nihr.ac.uk/glossary/?letter=F&postcategory=-1). In other words, a feasibility study is one that has not collected data, but has a study design and statistical support for that design. The study the authors have carried out might be better described as a “pilot trial” that shows positive results. To avoid the difficulty with the multiple meanings of feasibility, I suggest that the authors carefully examine their manuscript and locate all instances of “feasibility”, then replace it another word that conveys their meaning, such as “practicability”.

Author response: We agree in this comment, and have now changed the word/wording of “feasibility” into “practicability”. This can be seen in the title, as well as in the aim (line 78, page 5) and in the conclusion (line 48, page 3 (abstract), and line 307, page 20).

#2 The other important point the reviewer makes is that they have not done a power calculation to estimate how many subjects they need to claim that the results are statistically significant. A power calculation may not be possible without the data they collected, suggesting another reason to call this a pilot trial.

Author response: In the PASC trial protocol, a power calculation was performed to calculate the numbers needed per cluster. The initial numbers were 50 patients per cluster per month. We estimated that 50 patients per cluster would be sufficient to investigate how feasible this was, however, the COVID-19 pandemic situation influence on the numbers of patients recruited. We have provided the rationale for the sample size under Material and methods section, pages 6, lines 84-87: “Based on the power calculation in the PASC trail protocol, which was performed to calculate the numbers of elective surgical patients needed per cluster in the larger SW-CRCT, a need of 300 elective patients was estimated.”

Response to the Reviewer’s Comments

Reviewer #1: This study is presented as a feasibility study, however it doesn't seem to have actually been intended to assess feasibility of an intervention, as the objectives are presented as "investigating whether self-screening for nutritional risk factors prior to surgery in a home setting will increase the amount of patients screened for nutritional risk, and secondly, to compare their screening results with healthcare professionals..." (lines 26 to 29). Going by this, the objective of this study is to assess effectiveness of an intervention. The authors should be clearer about what the study was intending to achieve as the title, objectives in the abstract, and objectives at the end of the methods section are at odds.

If the authors did indeed aim to "investigate whether self screening... will increase the amount of patients screened...", then there are a number of aspects that need to be clearly reported, including the outcomes of the study and how they were measured, and also compared across groups. However, if the study aimed to investigate feasibility as stated in the title and in line 75, then the authors need to clearly describe what the feasibility endpoints are, ie. what outcome would they need to observe to conclude that self-screening was feasible. Additionally, the authors should describe how they determined the appropriate size of the study - a reader cannot tell whether the 215 patients who participated were sufficient to address the objectives of the study.

Author response: We agree in these comments, which is also commented by the Editor. Please see “Author response” to the Editor’s comments.

---

## [Decision Letter · Decision Letter 1]

4 Sep 2023

PONE-D-22-30022R1Practicability of self-screening for “at risk of malnutrition” among surgical patients’ in a home setting (ClinicalTrails.gov: NCT03105713)PLOS ONE

Dear Dr. Skeie,

Thank you for submitting your manuscript to PLOS ONE. After careful consideration, we feel that it has merit but does not fully meet PLOS ONE’s publication criteria as it currently stands. Therefore, we invite you to submit a revised version of the manuscript that addresses the points raised during the review process.

Your manuscript has been evaluated by four new reviewers, and their comments are appended below and in the attached document.

The reviewers have provided a range of comments regarding your manuscript, particularly regarding the study design and its communication in the text, the statistical methodology, and discussion of the results. Please ensure your address each of the reviewers' comments when revising your manuscript file.

Please also consider further the comments from Reviewer 1 of the previous round of peer review and assess whether the revision of 'feasible' to 'practicable' is a meaningful difference, and whether this study is assessing feasibility/practicability or effectiveness and reliability. Please make any revisions necessary to clarify this matter.

We look forward to receiving your revised manuscript.

Kind regards,

Hugh Cowley

Staff Editor

PLOS ONE

Reviewers' comments:

Reviewer's Responses to Questions

**Comments to the Author**

1. If the authors have adequately addressed your comments raised in a previous round of review and you feel that this manuscript is now acceptable for publication, you may indicate that here to bypass the “Comments to the Author” section, enter your conflict of interest statement in the “Confidential to Editor” section, and submit your "Accept" recommendation.

Reviewer #2: All comments have been addressed

Reviewer #3: (No Response)

Reviewer #4: (No Response)

Reviewer #5: (No Response)

2. Is the manuscript technically sound, and do the data support the conclusions?

Reviewer #2: Yes

Reviewer #3: Partly

Reviewer #4: Partly

Reviewer #5: Yes

3. Has the statistical analysis been performed appropriately and rigorously? 

Reviewer #2: Yes

Reviewer #3: No

Reviewer #4: Yes

Reviewer #5: Yes

4. Have the authors made all data underlying the findings in their manuscript fully available?

Reviewer #2: Yes

Reviewer #3: Yes

Reviewer #4: Yes

Reviewer #5: Yes

5. Is the manuscript presented in an intelligible fashion and written in standard English?

Reviewer #2: Yes

Reviewer #3: Yes

Reviewer #4: Yes

Reviewer #5: Yes

6. Review Comments to the Author

Reviewer #2: Generally, the content in the manuscript is well described.

Please make a minor technical amendment to the manuscript before finalised submission. Insert the comment in the attached file.

Reviewer #3: This manuscript aims to investigate the important question how we can improve access to nutrition screening preoperatively to identify patients at risk of malnutrition and therefore at increased risk of postoperative complications. The manuscript contains some interesting data but needs major revisions to enable interpretation of the findings.

Abstract:

Line 28 - amount should be number of patients

Line 45 - how many patients had both an healthcare professional and a self screening completed? This is the most important number and should be reported. What was the time difference between the screening tools being completed?

Line 47 - 48 - the conclusion does not reflect the results section. The agreement is stated as "poor".

Methods:

Line 87 - how was this sample size calculated? Please include further details.

Line 96 - 98 - were patients recruited while an inpatient or at an outpatient clinic appointment? It is not clear. Why did health professionals not complete the nutrition screening at this point in time rather than when patients were admitted for surgery?

Please add further details about the NRS 2002 questions and timing of screening.

Lines 105 to 107 - how did calculate BMI? Was this based on self-reported weight and height? Did they have to use tables on paper to work this out? Please include the exact screening tool questions that were used in this study in a table or figure or supplementary information.

Lines 113- 117. It is unclear if the scoring system for "at risk of malnutrition" was changed given patients did not complete all questions. Please include a comment about whether this is an acceptable use of the screening tool.

Section starting Line 101. When did patients complete the tool? If a patient completes the tool 12 weeks prior to the health professional completing the tool the inter-rater agreement cannot be compared b/c they are not assessing the same state. It is likely that changes in weight and food intake will occur during this period.

Line 119 - 120 - has this question been used in previous studies? Please provide evidence to support it's inclusion or at least justify why is has been included.

Statistical methods:

This section does not detail which kappa measure of agreement was used. Please review accordingly. Assumption 5 of Cohen's kappa was not met - "Assumption #5: The same two raters are used to judge all observations (e.g., patients). This has been referred to as having fixed or unique raters. If different raters were used for each observation (e.g., patient), Cohen's kappa is not the appropriate test to use. However, in this latter case, you could use Fleiss' kappa instead, which allows randomly chosen raters for each observation (e.g., patient)."

Results:

Line 169 - were the patients who believed they were at risk of malnutrition the same patients who were identified as at risk of malnutrition on the NRS2002? This is important to know.

Line 171-172 "All these patients were screened after surgery". The use of the screening tool after surgery is not detailed in the methods section. Please make it clear in the. methods section when HPs and patients completed the screening tool.

Line 192 - be aware over interpreting the statistics. The number of patients identified at risk of malnutrition was only 5% of the population and the agreement statistic is based on a small sample of 9 versus 11 screening tool results.

Discussion

Line 230 - this needs a reference please.

Line 255 - it is not clear what the "high prevelance indices" is referring to.

Line 260 - 262 - these results are not clearly presented in the results section. Please revise accordingly.

Line 264 - please add further details about the time difference to the results section. This is very important as timing of risk of malnutrition prior to surgery dictates the type of intervention that could be provided to reduce risk preoperatively.

Line 278 to 279 - yes very important point!

Line 284 - the time delay here is also important. For example, screening by health professionals at the time of booking surgery date would be a more accurate method to compare with self-screening at time of booking.

Reviewer #4: I reviewed the article "Practicability of self-screening for "at risk of malnutrition" among surgical patients in a home setting.” The study's primary purpose is to investigate and compare the agreement rates between preoperatively measured self-reported malnutrition screening at home by patients and at the hospital by nurses. Although the study has a good design and presentation, some major methodological points make the study lower quality.

1- Missing data when choosing the patients in the study is much more than expected, which may cause a bias in the analyses.

2- The hypothesis and purpose of the study needed to be well documented in the paper. I could not understand what this work would add to this area. Preoperatively self-measured malnutrition rate should be similar to measurements in the hospital. However, kappa analysis results show shallow coefficient scores.

Best regards,

Reviewer #5: ABSTRACT

Results – might be useful to include the numbers screened in hospital prior to surgery, I got the impression that those screened pre surgery was more important than post.

Conclusions – might also be useful to mention here that hospital screening for malnutrition risk doesn’t always occur before surgery, so limiting opportunity to treat and better prepare for surgery.

MAIN TEXT

Introduction (line 65) - font is different

Statistics – state what is an adequate level of agreement for this study. See: https://pubmed.ncbi.nlm.nih.gov/23092060/

Results - I think that there is maybe still a bit of confusion over what the study set out to achieve, more clarity is needed. If main aim is to assess if self-screening increases amount of patients screened for malnutrition risk prior to surgery, then this should be stated as a main outcome. Results should open with and highlight the number(%) out of 215 that self-screened prior to surgery compared to the number(%) out of 215 that were screened in hospital prior to surgery. Currently this information reads more like descriptive stats rather than the main outcome and there is no direct comparison between numbers self-screened and numbers screened in hospital prior to surgery. This would allow you state that ‘self-screening improved malnutrition screening by X% compared to hospital screening prior to surgery’

Limitations – another limitation could be that there was a low number of patients who were identified as at risk of malnutrition.

Discussion – I felt that this started quite negatively by highlighting the main limitation of this study (missing screenings at hospital) – suggest to move this further down in the discussion. You could instead open more positively – this study highlights the importance of self-screening at home and demonstrates that self-screening is possible and increases numbers of patients screened for malnutrition risk prior to surgery. I think this should help to make your primary outcome come across better as currently level of agreement (secondary outcome) overshadows the main outcome.

7. PLOS authors have the option to publish the peer review history of their article (what does this mean?). If published, this will include your full peer review and any attached files.

Reviewer #2: No

Reviewer #3: No

Reviewer #4: No

Reviewer #5: **Yes: **Debra Jones

---

## [Author Response · Author response to Decision Letter 1]

2 Nov 2023

Dear journal staff, 

We state as requested by the Journal Requirements the following: “This does not alter our adherence to PLOS ONE policies on sharing data and materials.”. The Journal Requirements also requested captions for our Supporting Information files which can now be found in the end of the manuscript. One of these also include an anonymized dataset. 

Kind regards, 

Randi J Tangvik

---

## [Decision Letter · Decision Letter 2]

29 Nov 2023

PONE-D-22-30022R2Is self-screening for 'at risk of malnutrition' feasible in a home setting? (ClinicalTrails.gov: NCT03105713)PLOS ONE

Dear Dr. Tangvik,

Thank you for submitting your manuscript to PLOS ONE. After careful consideration, we feel that it has merit but does not fully meet PLOS ONE’s publication criteria as it currently stands. Therefore, we invite you to submit a revised version of the manuscript that addresses the points raised during the review process.

The manuscript has been evaluated by three reviewers, and their comments are available below.

The reviewers have raised a number of concerns that need attention, including reference to comments provided at the last round of revision to which no response was provided. They request additional information on methodological aspects of the study and revisions to the wording used to increase clarity

Could you please revise the manuscript to carefully address the concerns raised?

We look forward to receiving your revised manuscript.

Kind regards,

Jennifer Tucker, PhD

Staff Editor

PLOS ONE

Additional Editor Comments (if provided):

Please ensure that you provide a detailed response to reviewers with your revised submission, addressing each reviewer comment point by point to allow for assessment of whether their requests have been completed.

Reviewers' comments:

Reviewer's Responses to Questions

**Comments to the Author**

1. If the authors have adequately addressed your comments raised in a previous round of review and you feel that this manuscript is now acceptable for publication, you may indicate that here to bypass the “Comments to the Author” section, enter your conflict of interest statement in the “Confidential to Editor” section, and submit your "Accept" recommendation.

Reviewer #3: All comments have been addressed

Reviewer #4: (No Response)

Reviewer #5: (No Response)

2. Is the manuscript technically sound, and do the data support the conclusions?

Reviewer #3: Partly

Reviewer #4: Partly

Reviewer #5: Yes

3. Has the statistical analysis been performed appropriately and rigorously? 

Reviewer #3: No

Reviewer #4: (No Response)

Reviewer #5: Yes

4. Have the authors made all data underlying the findings in their manuscript fully available?

Reviewer #3: Yes

Reviewer #4: (No Response)

Reviewer #5: Yes

5. Is the manuscript presented in an intelligible fashion and written in standard English?

Reviewer #3: Yes

Reviewer #4: (No Response)

Reviewer #5: No

6. Review Comments to the Author

Reviewer #3: Please see attachment that includes specific comments and recommendations based on the author review.

Reviewer #4: I could not see any responses to my comments in the response documents. Therefore, I still think there is a significant methodological and hypothetical concern with this paper,

Best regards,

Reviewer #5: I have attached my comments - I have suggested quite a few edits to language as I don't feel that it makes sense in places or that its particularly clear.

7. PLOS authors have the option to publish the peer review history of their article (what does this mean?). If published, this will include your full peer review and any attached files.

Reviewer #3: No

Reviewer #4: No

Reviewer #5: **Yes: **Debra Jones

---

## [Author Response · Author response to Decision Letter 2]

15 Jan 2024

Dear Editor Hugh Cowley, Staff Editor Jennifer Tucker, and the reviewers

PLOS ONE

Thank you for valuable feedback. We apologize that some corrections were not visible last time and have ensured that they all are this time. The methodology is revised and thoroughly explained in the revised Figure 1, 2 and 3, in the Material and Methods section and the Result section. Also, the Discussion is revised, and the Conclusion is rewritten. We hope you will find this satisfactory. 

We hope that the revised manuscript has addressed the reviewers’ suggestions satisfactorily and is now at a level that can be deemed acceptable for publication in PLOS ONE. 

Yours sincerely, 

Randi J Tangvik, on behalf of all the authors

---

## [Decision Letter · Decision Letter 3]

8 Feb 2024

Is self-screening for 'at risk of malnutrition' feasible in a home setting? (ClinicalTrails.gov: NCT03105713)

PONE-D-22-30022R3

Dear Randi Tangvik,

We’re pleased to inform you that your manuscript has been judged scientifically suitable for publication and will be formally accepted for publication once it meets all outstanding technical requirements.

Kind regards,

Mohammed Hasen Badeso, MPH in Field Epidemiology

Academic Editor

PLOS ONE

Additional Editor Comments (optional):

Reviewers' comments:

Reviewer's Responses to Questions

**Comments to the Author**

1. If the authors have adequately addressed your comments raised in a previous round of review and you feel that this manuscript is now acceptable for publication, you may indicate that here to bypass the “Comments to the Author” section, enter your conflict of interest statement in the “Confidential to Editor” section, and submit your "Accept" recommendation.

Reviewer #1: (No Response)

Reviewer #4: All comments have been addressed

2. Is the manuscript technically sound, and do the data support the conclusions?

Reviewer #1: Yes

Reviewer #4: Yes

3. Has the statistical analysis been performed appropriately and rigorously? 

Reviewer #1: Yes

Reviewer #4: Yes

4. Have the authors made all data underlying the findings in their manuscript fully available?

Reviewer #1: Yes

Reviewer #4: Yes

5. Is the manuscript presented in an intelligible fashion and written in standard English?

Reviewer #1: Yes

Reviewer #4: Yes

6. Review Comments to the Author

Reviewer #1: This is a generally well-described study, I have only a few minor comments.

- In line 93 do you mean 'trial' instead of 'trail'?

- Also in the same part, I think a bit more detail about the sample size of the overarching study might be useful, for example, what size of effect or difference for what outcome with how much power at what level of significance and what ICC for clustering did the study deem 300 participants in six clusters to be sufficient.

- In line 152 you likely mean 'estimated' instead of 'investigated', and in line 149 'used' instead of 'applied'.

Reviewer #4: Thank you for your appropriate revisions. The last version of the study seems to be ready to be accepted for publication.

Best regards,

7. PLOS authors have the option to publish the peer review history of their article (what does this mean?). If published, this will include your full peer review and any attached files.

Reviewer #1: No

Reviewer #4: No

---

## [Editor Report · Acceptance letter]

23 Feb 2024

PONE-D-22-30022R3 

PLOS ONE

Dear Dr. Tangvik, 

I'm pleased to inform you that your manuscript has been deemed suitable for publication in PLOS ONE. Congratulations! Your manuscript is now being handed over to our production team.

Kind regards, 

on behalf of

Mr Mohammed Hasen Badeso 

Academic Editor

PLOS ONE